# Verified Learning for Compiler Optimization: An LLM-Guided Architecture with Formal Control

## Abstract

Compiler optimizations traditionally rely on handcrafted heuristics that often fail to generalize across programs and architectures. We investigate whether large language models can participate in compiler optimization through a verification-centered systems architecture that couples generative rewriting with formal equivalence checking. Using lazification in LLVM IR as a case study, we fine-tune a code-centric LLM on transformations produced by Wyvern and embed Alive2 into a feedback loop that enforces semantic preservation for every generated rewrite. Correctness is enforced externally as a runtime control layer rather than learned implicitly.

During inference, candidate transformations are symbolically validated and regenerated when necessary, ensuring accepted rewrites satisfy formal constraints. On the LLVM test suite, the fine-tuned model reproduces core optimization behaviors while applying fewer transformations overall. Although Wyvern remains faster on most benchmarks, 9.8% achieve comparable or improved runtime under the learned system, with no semantic violations observed. Verification overhead remains bounded and convergence stable. These results demonstrate that generative AI components can be safely integrated into compiler pipelines through deterministic validation and structured feedback, offering a scalable architectural pattern for trustworthy AI-driven software infrastructure.

## CCS Concepts

• **Software and its engineering** → **Compilers**.

## Keywords

Verified Learning, Compiler Optimization, Large Language Models, Formal Verification, AI-Assisted Software Engineering

**ACM Reference Format:**

Anonymous Author(s). 2018. Verified Learning for Compiler Optimization: An LLM-Guided Architecture with Formal Control. In *Proceedings of Make sure to enter the correct conference title from your rights confirmation email (Conference acronym 'XX)*. ACM, New York, NY, USA, 9 pages. https://doi.org/XXXXXXX.XXXXXXX

## 1 Introduction

Compiler optimizations have traditionally relied on handcrafted heuristics that encode expert intuition about program behavior. Although these rules have driven decades of progress, they often

generalize poorly across architectures and workloads, leading to brittle cost models that require repeated platform-specific tuning [7, 13]. This limitation has motivated data-driven optimization, where models learn profitability patterns directly from program data. Early systems such as Milepost GCC [7] and AutoPhase [10] demonstrated that supervised and reinforcement learning can outperform static heuristics for flag selection and phase ordering. Subsequent work, including Ithemal [13] and NeuroVectorizer [9], replaced analytical cost models with learned predictors. More recently, MLGO [19] and the LLM Compiler project [5] trained large models over IR and assembly corpora to support tasks such as inlining, register allocation, and IR synthesis. Despite this progress, most learning-driven compilers optimize runtime or code size without offering transformation-level correctness guarantees. In production toolchains, even rare semantic violations are unacceptable, limiting how aggressively learned optimizers can be deployed. This tension exposes a broader software engineering challenge: how can generative AI components be embedded into critical infrastructure systems without compromising reliability or formal correctness?

We argue that the core challenge for AI-powered compilation is not predictive accuracy alone, but the integration of learning with formal guarantees. To examine this tension, we study lazification, a transformation that converts eager call-by-value evaluation into deferred call-by-need semantics [2]. Lazification can remove unnecessary computations when arguments are rarely used, yet it can introduce overhead through thunks and memoization when access frequency is high [2]. Profitability therefore depends on dynamic usage patterns and control-flow context, making it a demanding setting for heuristic reasoning. Systems such as Wyvern rely on manually tuned rules derived from profiling, which may mispredict on unseen programs and introduce regressions [7, 10, 13]. Rather than replacing compiler logic with unconstrained generative models, we propose a verification-centered workflow in which learned models suggest candidate transformations that are executed only after mechanical proof of equivalence. In this formulation, the compiler remains the authority on correctness, while the LLM serves as a proposal engine within a formally governed pipeline.

Our framework combines the adaptability of large language models with the formal rigor of verified compilation. We fine-tune a code-centric LLM on function-level transformations produced by Wyvern, treating those transformations as supervision for profitable lazification patterns. At inference time, the LLM proposes lazification candidates on LLVM IR, and each candidate rewrite is formally validated for semantic equivalence using Alive2 [11]. Transformations are applied only when the verifier proves preservation of program behavior; otherwise, diagnostic information is returned and regeneration is triggered. This verification-guided loop bridges learning-based adaptation with compiler-level assurance [8]. Importantly, correctness is not learned by the model but enforced externally by the verifier. The LLM is trained solely on

transformation pairs and does not receive verification feedback during fine-tuning; formal validation is introduced only at inference time, where it functions as an external control mechanism over generated rewrites. This design separates transformation generation from correctness enforcement while still enabling verifier-guided refinement. Architecturally, this separation converts verification into a runtime governance layer that constrains generative behavior without suppressing adaptivity. As a result, optimization behavior can be learned adaptively, yet semantic correctness remains guaranteed independently of model prediction accuracy.

This paper makes three contributions. First, we introduce a verified learning architecture that couples LLM-based rewrite proposal with automated semantic validation at the IR level. Second, we construct a lazification-focused dataset and evaluation protocol that separate transformation applicability from correctness. Third, we empirically analyze how verification constraints influence optimization behavior and reliability outcomes. Evaluation on the LLVM test suite shows that the fine-tuned model closely reproduces Wyvern's optimization patterns while applying fewer lazifications overall. Approximately 76% of benchmarks exhibit reduced transformation scope and runtime slowdowns, whereas about 10% achieve comparable or improved performance. No semantic violations or over-lazification cases were observed. Compared with a non-finetuned baseline, the model achieves a 2.5% runtime reduction and 9.5% smaller binaries. Structural analysis indicates a preference for conservative rewrites with increased load and store activity, reflecting a safety-oriented style. Taken together, these results demonstrate that formally constrained generative components can be integrated into production compiler pipelines in a predictable and verifiable manner, offering a reusable architectural pattern for trustworthy AI-assisted software transformation systems.

## 2 Related Work

The most directly related work is Wyvern, introduced in Lazy Evaluation for the Lazy [2]. Wyvern demonstrated lazification as a compiler optimization for C, transforming call-by-value into call-by-need through heuristic and profiling-guided decisions. While effective on tuned benchmarks, this approach exposed limitations of heuristic design: rules tailored to specific workloads rarely generalize, and profiling introduces compilation overhead without guaranteeing gains. Our work builds on this foundation but shifts the question from heuristic design to architectural design: can learned models replace handcrafted decision logic while correctness is enforced independently? Unlike Wyvern, which encodes optimization knowledge through manually engineered rules, we learn rewrite proposals from verified transformations as supervision and rely on an external validation layer to guarantee semantic correctness.

The broader effort to replace compiler heuristics with learning dates back to Milepost GCC [7], which showed that supervised models could outperform static rules for flag selection. AutoPhase [10] applied reinforcement learning to pass ordering, and MLGO [17] deployed learned policies in LLVM for inlining and register allocation. Complementary efforts targeted AI-driven optimization for deep learning workloads [16]. Collectively, these systems demonstrated that machine learning can replace handcrafted cost models. However, they frame optimization primarily as a performance-selection problem rather than a semantics-sensitive transformation problem. In contrast, our focus is on transformations whose validity depends on formal semantic equivalence, introducing fundamentally different constraints on learned optimizers.

Subsequent work explored richer structural representations of code. Ithemal [13] replaced analytical throughput models with neural predictors of basic-block performance, while inst2vec [1] and ProGraML [3] introduced graph-based embeddings over IR control and data flow. Reinforcement-learning approaches such as NeuroVectorizer [9] and RL4ReAl [18] extended these ideas to loop vectorization and register allocation. These works show that deep models can internalize compiler structure and guide fine-grained optimizations. Yet they assume semantic soundness of the transformation space and do not enforce equivalence for each generated rewrite. Our work retains structural learning but introduces a verification-driven acceptance mechanism that constrains transformation application independently of model confidence.

A complementary line of research centers on correctness. Souper [14] and STOKE [15] exemplify superoptimization, searching for improved code sequences and validating them through synthesis or probabilistic checks. Alive [12] and Alive2 [11] formalized equivalence checking within LLVM, enabling automated validation of candidate rewrites. These systems guarantee correctness but lack adaptivity in deciding when transformations are profitable. Our framework unifies these directions by embedding Alive2 into a learning-guided pipeline, turning verification from a post-hoc filter into a continuous control mechanism within optimization.

More recently, large language models have entered compiler optimization. The LLM Compiler project [5] trained foundation models on LLVM IR and assembly for general optimization tasks, while follow-up work explored compiler-guided feedback [8], optimization reasoning [4], and reinforcement-based refinement [19]. CompilerDream [6] framed LLMs as world models for optimization planning. These studies demonstrate that LLMs can learn meaningful program abstractions. However, they evaluate optimization primarily through empirical performance metrics and do not couple generation with formal equivalence enforcement for semantics-sensitive rewrites. Our work explicitly studies how verification constraints shape learned transformation behavior.

Hybrid systems that combine deterministic compiler components with learned reasoning are also emerging. Zhang et al. [20] showed how LLMs and compilers can collaborate for Python refactoring, blending symbolic precision with adaptive generalization. Our design follows a similar hybrid philosophy but applies it to IR-level compiler optimization: deterministic components such as Wyvern and Alive2 provide supervision and correctness guarantees, while the fine-tuned LLM supplies contextual adaptivity. The key distinction is that in our system, the verifier serves as the ultimate decision authority, ensuring that semantic correctness is independent of model behavior.

In summary, prior research has explored heuristics, learning-guided optimization, structural code modeling, verified superoptimization, and LLM-based reasoning largely in isolation. Our work unifies these directions through a verified learning architecture that tightly couples adaptive LLM-based transformation generation with formal semantic validation, advancing a systems-oriented approach to trustworthy compiler optimization.

## 3  Approach

Our approach instantiates verified learning, a framework that couples data-driven adaptivity with formal correctness guarantees. We fine-tune a large language model to decide when eager call-by-value computations in C can be transformed into deferred call-by-need semantics. The system integrates three components: the Wyvern optimizer, which provides verified training examples; the Meta Compiler LLM, which generalizes these transformation patterns to unseen code; and the Alive2 verifier, which proves semantic equivalence of each generated rewrite. Combined, they form a closed-loop pipeline that learns from verified transformations, proposes new candidates, and validates them symbolically before acceptance. Unlike traditional learning-based compiler systems where correctness is an implicit expectation, our design treats correctness as an externally enforced invariant, allowing the model to explore optimization opportunities without risking semantic violations.

### 3.1  Problem Formulation

At its core, the problem is a conditional transformation task: determining, for each function, whether applying *lazification* improves performance without violating semantic equivalence. Given a function $f$ in LLVM Intermediate Representation (IR), the goal is to produce a transformed version $f'$ that preserves program behavior but converts eager *call-by-value* evaluation into deferred *call-by-need* execution only when profitable.

$$H : \mathcal{F} \to \{\text{transform, skip}\},$$

Traditional compilers rely on hand-crafted heuristics, often tuned by profiling, to make this decision. We instead learn a transformation proposal mapping using a decoder-only large language model trained in a sequence-to-sequence fashion:

$$f_{\text{LLM}} : \text{IR}_{\text{unopt}} \to \text{IR}_{\text{lazified}},$$

where $\text{IR}_{\text{unopt}}$ denotes unoptimized LLVM IR generated by `Clang`, and $\text{IR}_{\text{lazified}}$ is the LLM's selectively transformed version.

### 3.2  System Overview

Figure 1 outlines the architecture of our LLM-guided lazification framework, which operates in two phases: *training* and *inference*. The pipeline integrates within the compiler toolchain, ensuring that every learned transformation remains both semantically valid and performance-sensitive. By embedding verification directly within the transformation lifecycle, the system ensures that the compiler remains the final authority on correctness while the LLM acts as a proposal generator rather than a decision oracle.

During the *training phase*, C source files are first compiled to unoptimized LLVM Intermediate Representation (IR) using `Clang-14`. The resulting IR is then processed by `Wyvern`, which acts as a heuristic oracle to produce the corresponding lazified versions. Unlike traditional filtered datasets, we retain both profitable and unprofitable transformations so that the model can observe the full decision space, including cases where lazification degrades performance. This enables the Meta Compiler LLM to learn the structural and usage patterns that distinguish safe, profitable transformations from harmful ones. Retaining unsuccessful transformations exposes the model to cases where lazification degrades performance, helping it

learn when the optimization should be avoided rather than simply memorizing successful rewrite patterns.

In the *inference phase*, the process is reversed. A new program is compiled into LLVM IR and provided to the fine-tuned model, which generates candidate lazifications. Each candidate is subsequently validated by `Alive2` to ensure semantic equivalence before code generation. When verification fails or compilation errors arise, diagnostic traces are fed back to the model for self-correction. This feedback loop tightly couples generative adaptation with formal reasoning, allowing the system to evolve its optimization behavior while preserving provable correctness. This design converts verification failures into structured feedback that guides iterative refinement while preserving correctness guarantees.

### 3.3  Research Questions

To structure our investigation, we formulate the following research questions:

- **RQ1:** How effectively can a large language model reproduce the optimization quality of a verified compiler heuristic?
- **RQ2:** What structural optimization behaviors emerge when the model replaces handcrafted rules?
- **RQ3:** To what extent do formal verification and feedback loops ensure correctness in learned compiler transformations?

These questions ground the evaluation in the core objectives of verified learning for compiler optimization, focusing on performance, structural behavior, and correctness guarantees. RQ1 examines behavioral fidelity relative to expert heuristics, RQ2 analyzes structural transformation patterns that emerge without handcrafted rules, and RQ3 evaluates whether formal verification enables stable convergence under generative control. Collectively, they frame the study as both an empirical assessment and an architectural investigation of reliable AI-assisted compiler transformation.

### 3.4  Dataset

To train and evaluate our model, we curated a function-level dataset from a diverse suite of real-world open-source C libraries. Our objective was to capture authentic optimization patterns that arise in production software rather than relying on synthetic microbenchmarks. Real-world code exposes the model to realistic control-flow complexity, memory-access behavior, and optimization trade-offs rarely present in isolated kernels.

The training corpus comprises eight widely used C projects: `cJSON`, `libevent`, `libtommath`, `sqlite`, `zlib`, `libconfuse`, `libgit2`, and `lz4`. These libraries span domains ranging from parsing utilities and numerical routines to embedded databases and compression engines, ensuring representative control-flow and performance-critical idioms relevant to lazification. Each example corresponds to a single function. Source files are compiled to LLVM IR using `clang -S -emit-llvm -Xclang -disable-O0-optnone`, and the resulting IR is processed by Wyvern, which heuristically applies lazification when profitable. Crucially, we retain both transformed and untransformed functions, exposing the model to the full decision boundary of applicability rather than only successful rewrites. Training therefore spans all transformation outcomes, enabling the LLM to learn when to apply and when to avoid lazification, which

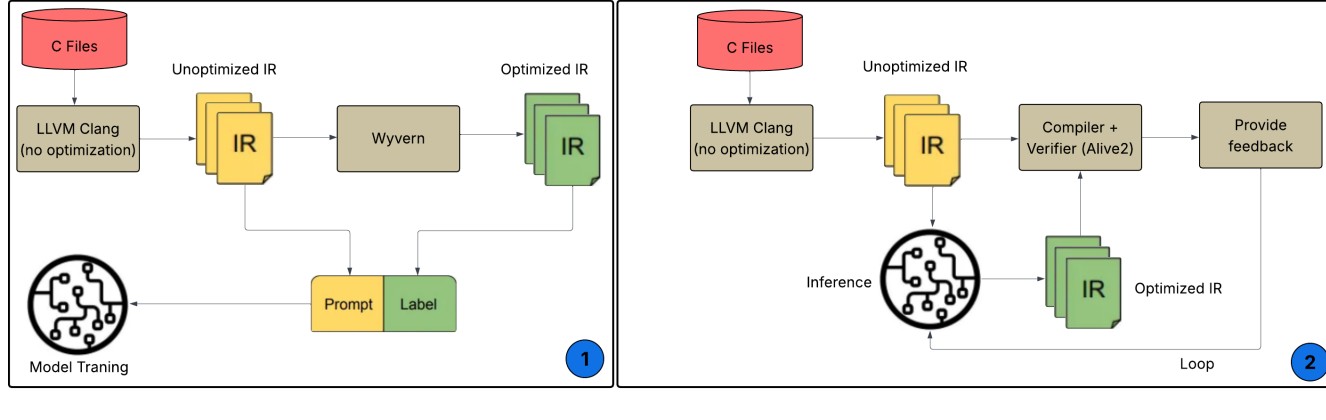

**Figure 1: Two-phase LLM-guided lazification framework. During training, LLM IR from C programs is paired with Wyvern-optimized IR to fine-tune the model. During inference, the model generates lazified IR verified by Alive2 before compilation.**

**Table 1: We report lines of code (LOC), function counts, lazifiable functions, and average IR size for regular and lazified code.**

| Codebase | LOC | #Functions | Avg. IR LOC/Function | #Lazyfiable Functions | Avg. IR LOC/Lazyfiable |
|---|---|---|---|---|---|
| cJSON | 34,886 | 208 | 73.38 | 54 | 52.02 |
| LIBEVENT | 117,416 | 30 | 136.27 | 3 | 98.85 |
| LIBTOMMATH | 30,906 | 351 | 139.46 | 40 | 105.36 |
| SQLITE | 1,673,688 | 1,033 | 133.88 | 348 | 94.31 |
| ZLIB | 91,017 | 307 | 185.46 | 77 | 135.65 |
| LIBCONFUSE | 17,342 | 40 | 197.61 | 1 | 182.61 |
| LIBGIT2 | 33,147 | 174 | 259.21 | 69 | 175.24 |
| LZ4 | 41,547 | 164 | 659.55 | 23 | 324.42 |
| TOTAL (TRAINING) | 2,039,949 | 2,307 | 223.10 | 615 | 146.06 |
| **LLVM-TEST-SUITE (TESTING)** | **1,943,580** | **8,422** | **53.78** | - | - |

reduces bias toward transformation-positive samples and encourages context-sensitive policies. To evaluate generalization, we hold out the `llvm-test-suite` as a dedicated test corpus containing over 8,000 C functions from diverse system and application benchmarks. These programs are compiled using the same pipeline with no overlap with the training libraries, ensuring out-of-distribution evaluation rather than project-specific memorization.

Table 1 summarizes both corpora. The training set contains 2,307 functions across 2.04M lines of code, including 615 lazified by Wyvern. Lazifiable functions are longer and structurally richer (146 IR lines on average), reflecting the complexity of profitable transformations. While `sqlite` contributes the largest share, all libraries add functional diversity. The `llvm-test-suite` adds 1.94M lines of unseen code, enabling systematic analysis of how learned transformation behavior adapts across varying program structures and optimization opportunities.

### 3.5 Wyvern Optimizer Oracle

Wyvern serves as the verified heuristic oracle used to construct our training corpus. It performs *lazification* by selectively converting call-by-value parameters into deferred call-by-need computations using lightweight thunks and memoization. The transformation preserves semantic equivalence while potentially improving runtime performance. Each unoptimized LLVM IR function is analyzed by Wyvern's rule-based profitability model, which considers control flow, data dependencies, and parameter usage frequency. Parameters accessed conditionally or infrequently are flagged as lazifiable, triggering rewrites that insert thunk and memoization logic at use sites. The resulting verified transformation pairs ($IR_{unopt}$, $IR_{wyvern}$) provide supervision for training. Rather than treating Wyvern as an absolute oracle, we use its outputs as verified exemplars, encouraging the model to learn transferable decision features instead of reproducing fixed heuristic rules.

### 3.6 Meta Compiler LLM

At the core of the system is the *Meta Compiler LLM*, built on the Code LLaMA architecture. Pre-trained on compiler-relevant corpora including LLVM IR, assembly, and structured edits, the model is designed to reason over low-level control flow and memory operations. We fine-tune it using verified ($IR_{unopt}$, $IR_{wyvern}$) pairs, enabling the model to learn when lazification is beneficial and how

to apply it within semantic constraints. Unlike handcrafted heuristics, the fine-tuned model captures higher-order structural signals and generalizes to unseen program patterns. It therefore acts as a context-sensitive transformation synthesizer that complements deterministic analyses rather than replacing them.

### 3.7 Alive2-Guided Verification and Integration

To enforce semantic correctness, we integrate *Alive2* [11] into the inference loop. Alive2 symbolically compares the original and transformed LLVM IR using SMT solvers to verify equivalence under all inputs. Only rewrites that pass verification are accepted; failures trigger regeneration guided by diagnostic traces. This design positions verification as an inline control mechanism within the optimization pipeline rather than a post-hoc filter. By tightly coupling generative proposal with formal validation, the system enables adaptive optimization while preserving semantic guarantees.

### 4 Experimental Setup

All experiments are conducted in a controlled environment to ensure reproducibility and fair comparison across *Wyvern*, the unoptimized LLVM pipeline, and both variants of the *Meta Compiler LLM*. Compiler flags, random seeds, verifier configurations, dataset partitions, and evaluation metrics are fixed across runs to eliminate nondeterministic variation, ensuring that observed differences arise from model-driven transformations rather than environmental noise. Experiments run on an Intel Core i5-120U (12-core) system with 16 GB RAM under Ubuntu 22.04 LTS. LLVM 14 and Clang 14 generate unoptimized IR, while *Wyvern* is integrated as an LLVM plugin with Z3-backed SMT support. Fine-tuning and inference are performed on an NVIDIA H200 SXM GPU (141 GB VRAM) using CUDA 12.8 and PyTorch 2.8. Runtime measurements are collected on the CPU to ensure fair comparison with compiler baselines.

Training data consists of eight open-source C libraries using an 80/20 train-validation split. Evaluation is performed exclusively on the held-out `LLVM-test-suite`, and no evaluation functions are used during model selection or hyperparameter tuning. We evaluate both a pre-trained and a fine-tuned Code LLaMA model. Fine-tuning employs LoRA with an 8K token context window, a learning rate of $1 \times 10^{-5}$, batch size 2, and three epochs, using `bfloat16` precision without weight decay. These settings prioritize stability and parameter efficiency while preserving pretrained structural inductive biases. Evaluation measures semantic correctness via Alive2 equivalence checking, structural metrics relative to Wyvern transformations, and runtime and binary size deltas. Results are summarized using medians and bootstrap confidence intervals to capture variability across programs.

### 5 Results

We evaluate the proposed LLM-guided lazification framework across runtime efficiency, binary size, and semantic correctness. Results show that the fine-tuned model closely mirrors Wyvern's optimization quality, reproducing verified lazifications with high structural fidelity and stable runtime behavior. While the model applies fewer transformations overall, it maintains full semantic correctness, as validated by Alive2. Comparative analyses further reveal consistent improvements over the non–fine-tuned baseline, underscoring the

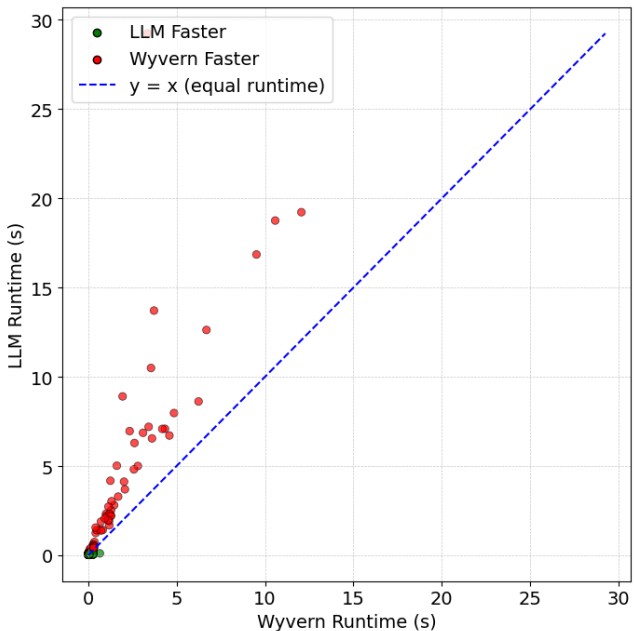

**Figure 2: Comparison of runtime between fine-tuned LLM and Wyvern across benchmarks.**

value of combining data-driven adaptivity with formal verification. Unless otherwise noted, runtime measurements report median execution time across repeated runs to reduce noise and reflect stable performance trends.

### 5.1 Optimization Fidelity and Runtime Efficiency

This section describes the quantitative relationship between the fine-tuned LLM and Wyvern across runtime, transformation scope, and binary size. Figure 2 presents a scatter plot comparing execution time of programs optimized by the fine-tuned LLM against Wyvern. Each point corresponds to a benchmark, with the x-axis denoting Wyvern runtime and the y-axis denoting LLM runtime. The blue dashed line marks the equality boundary ($y = x$). Most points cluster along or slightly above the diagonal, indicating that the fine-tuned LLM closely tracks Wyvern across a wide range of workloads. Approximately 90% of benchmarks lie above the equality line, showing that Wyvern executes faster in most cases, while about 10% fall below the line, where the LLM achieves modest runtime improvements. The tight clustering near $y = x$ confirms a consistent performance relationship rather than erratic variance and preserves relative performance ordering across benchmarks. Benchmarks under 1s cluster near the origin due to near-identical performance or measurement noise. A few bounded outliers exhibit larger gaps, typically arising from heavy argument reuse patterns where deferred evaluation introduces thunk overhead.

Figure 3 illustrates how differences in lazified functions relate to runtime outcomes. Each cross represents a benchmark, with Δ Functions (LLM – Wyvern) on the x-axis and Δ Runtime (%) on the y-axis.

Table 2: Summary of Runtime and Binary-Size Metrics.

| Metric | Wyvern Mean | LLM Mean | Δ (LLM – Wyvern) % | Wyvern Median | LLM Median | Benchmarks Improved (%) |
|---|---|---|---|---|---|---|
| Runtime (s) | 0.1104 | 0.2178 | +97.29 | 0.0331 | 0.0579 | 9.8 |
| Binary Size (MB) | 0.01549 | 0.01401 | −9.53 | 0.01518 | 0.01518 | 10.8 |

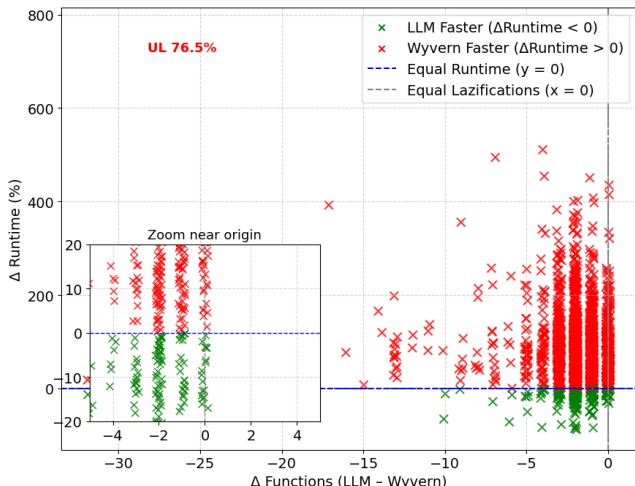

Figure 3: Relationship between lazified functions and runtime difference across benchmarks.

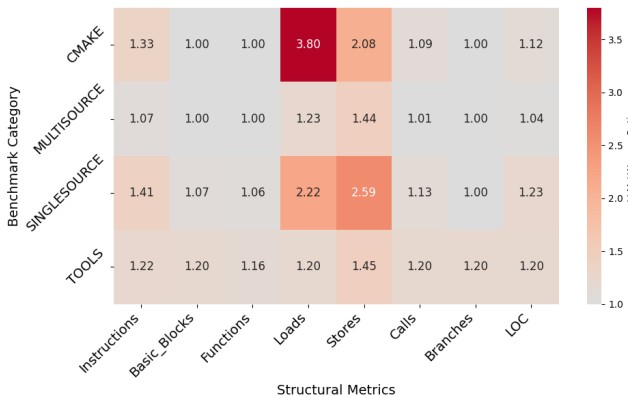

Figure 4: Heatmap of LLM-to-Wyvern ratios for key structural metrics across benchmark categories.

Red markers indicate cases where Wyvern is faster, and green markers denote LLM gains. The dashed lines mark equal lazification and equal runtime. The distribution is strongly left-skewed, indicating that the LLM generally applies the same or fewer lazifications than Wyvern. Roughly 76% of benchmarks fall in the upper-left quadrant, reflecting fewer transformations and slightly slower runtimes. About 9% appear in the lower-left quadrant, where fewer transformations still yield faster execution, demonstrating selective effectiveness. No benchmarks appear on the right side (Δ Functions > 0), confirming that the model does not over-lazify. The remaining ≈15% cluster near the axes, where both systems produce nearly identical results. This pattern indicates that verification-constrained learning biases the model toward under-transformation rather than speculative rewriting, thereby reducing regression risk.

Table 2 summarizes system-level metrics. The LLM's mean runtime is nearly double Wyvern's (+97%), though 9.8% of benchmarks run faster under the LLM. This gap is driven by a subset of slower benchmarks, while median runtimes remain sub-second for both systems with no severe regressions. The LLM produces smaller binaries on average (−9.5%), with 10.8% of benchmarks yielding more compact outputs, while medians remain identical. Bootstrap confidence intervals indicate bounded runtime variance, suggesting systematic differences rather than measurement noise.

## 5.2 Structural Optimization Behavior

Figure 4 illustrates how the fine-tuned LLM modifies program structure relative to Wyvern's handcrafted optimization rules. Each cell

represents the ratio between the LLM's and Wyvern's structural metrics, computed as

$$\text{Ratio} = \frac{\text{LLM metric value}}{\text{Wyvern metric value}},$$

where values above 1.0 indicate that the LLM produces more structural elements and values below 1.0 indicate fewer. We focus on IR-level statistics, including *Instructions*, *Basic Blocks*, *Functions*, *Loads*, *Stores*, *Calls*, *Branches*, and *Lines of Code (LOC)*, as these capture transformation scope, control-flow structure, and memory-operation density. All ratios are aggregated at the benchmark level using medians to reduce sensitivity to extreme IR expansions.

Across benchmark categories (*CMAKE*, *MULTISOURCE*, *SINGLE-SOURCE*, and *TOOLS*), the heatmap reveals a consistent and conservative pattern. The *ABI* category is excluded due to a single-file artifact that produces undefined ratios. The LLM emits roughly 1.3× as many *Instructions* in *CMAKE* and *SINGLESOURCE*, with modest increases in *Basic Blocks* (≈ 1.0–1.2×) and *Functions* (≈ 1.0–1.16×). The largest deviations occur in *Loads* (up to 3.8× in *CMAKE*) and *Stores* (up to 2.6× in *SINGLESOURCE*), reflecting cautious memory expansion from thunk and memoization scaffolding. *Branches* remain near 1.0, and *Calls* increase slightly (1.09–1.20×), indicating that control-flow topology is largely preserved.

Overall, *LOC* expands modestly (1.04–1.23×), yielding verbose yet semantically stable output. Crucially, this structural expansion does not correspond to runtime divergence, as shown in Section 2. Collectively, these results indicate that the LLM adopts a correctness-preserving optimization style shaped by verification constraints. Rather than minimizing structural metrics directly, the

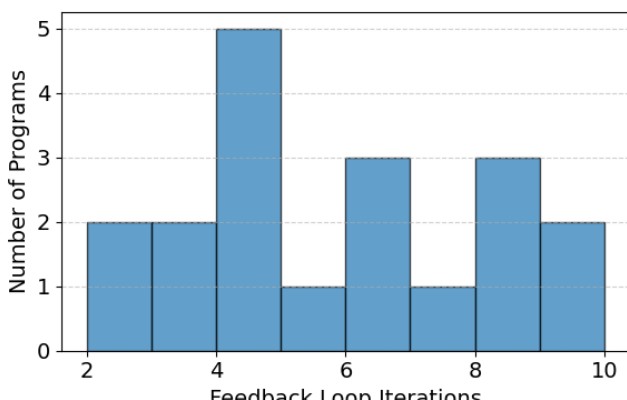

**Figure 5: Convergence of the feedback loop across 19 programs. Most stabilized within 4–6 iterations (mean=5.32, std=2.08), confirming efficient and stable correction.**

model operates within stable envelopes, favoring safety and interpretability over aggressive compaction. When coupled with formal verification, generative optimization gravitates toward under-transformation rather than speculative restructuring, resulting in predictable structural behavior across workloads.

### 5.3  Feedback-Loop Convergence

Figure 5 reports convergence statistics for the 19 programs that initially failed semantic or compilation checks. Among these, 89.5% exhibited only syntax errors, 10.5% contained both syntax and semantic errors, and none failed solely due to semantics, underscoring strong semantic alignment and the primarily syntactic fragility of generated IR. This pattern indicates that the model typically preserves semantic intent while occasionally violating surface-level IR constraints, remaining close to valid transformation boundaries rather than committing deep logical errors. The number of verification-guided iterations required for correction ranged from 2 to 9, with a mean of 5.32 and a standard deviation of 2.08. Most cases converged within four to six iterations, forming a concentrated central band, and none exhibited divergence or oscillation. No instance required manual intervention or fallback to heuristic rewriting, demonstrating that verifier-guided regeneration alone was sufficient for recovery. Each verification step incurs an average latency of 2.46 ms, so even in the worst case of nine iterations, cumulative validation overhead remains modest relative to compilation time. Collectively, these results indicate that the feedback loop exhibits stable, bounded convergence rather than open-ended search, supporting the case for embedding symbolic verification within generative compiler pipelines.

### 5.4  Impact of Fine-Tuning

To enable fair comparison between runtime and binary size, despite differences in units and scale, both metrics were normalized relative to the non-finetuned mean:

$$M_{\text{norm}} = \frac{M}{M_{\text{non-finetuned mean}}} \times 100.$$

**Table 3: Runtime and binary size of the fine-tuned model, normalized to the baseline. Values reflect relative change.**

| Metric | Non Finetuned | Finetuned | Relative Change |
|---|---|---|---|
| Runtime (s) | 100.0% | 97.5% | −2.5% |
| Binary Size (MB) | 100.0% | 90.5% | −9.5% |

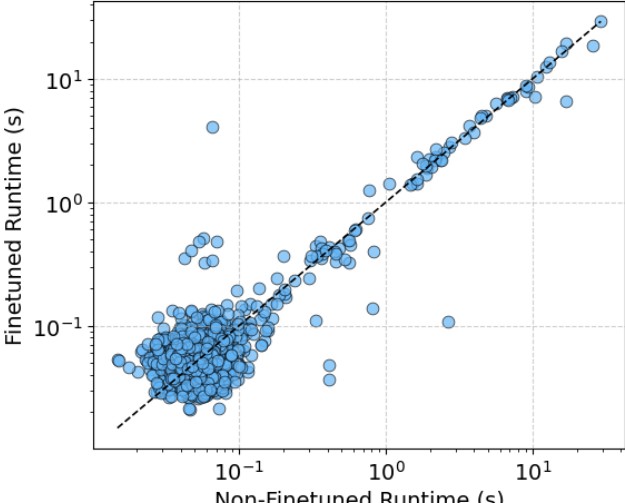

**Figure 6: Comparison of fine-tuned and non-finetuned model runtimes across benchmarks.**

This rescaling expresses each value as a percentage of the baseline, allowing compact comparison across metrics. All reported improvements are computed over the full benchmark suite to avoid selection bias, and medians exhibit similar directional trends, suggesting that the gains are not primarily driven by a small subset of outliers. As shown in Table 3, fine-tuning yields a 2.5% reduction in average runtime and a 9.5% reduction in binary size, indicating improved efficiency and compactness. These improvements, while modest in magnitude, are achieved without sacrificing formal correctness guarantees, underscoring that specialization on verified transformations enhances calibration rather than aggressiveness.

Figure 6 plots per-benchmark runtime on log–log axes for both models. Roughly 78% of benchmarks fall below the $y = x$ line, confirming consistent runtime improvements. Most points cluster near the diagonal, with only minor variance and few outliers, suggesting that fine-tuning preserves stability while generalizing optimization behaviors across the suite. The absence of extreme regressions further indicates bounded behavioral variance, aligning the model more closely with verified optimization boundaries rather than amplifying speculative transformations.

## 6  Discussion

This section interprets the results through the research questions, highlighting how verified learning influences optimization fidelity, structural behavior, and convergence stability. Beyond performance

comparison, the analysis examines how verification constraints shape compiler behavior. More broadly, the results show how generative AI components can be embedded within safety-critical software pipelines without relinquishing formal control.

**RQ1:** The fine-tuned model captures much of the verified baseline's optimization behavior, producing functionally correct, size-reduced binaries and modest runtime gains on a subset of benchmarks. Although it applies fewer transformations overall, optimizing a substantially smaller subset of functions than Wyvern, it consistently preserves semantic correctness under Alive2 validation. The absence of over-lazified code and the alignment of runtime trends suggest that the model learns transformation applicability patterns rather than producing arbitrary or unstable rewrites. Rather than replicating all heuristic decisions, the model learns applicability conditions and adopts a cautious transformation policy. From a systems perspective, this behavior demonstrates that constrained generative models can approximate expert-designed logic while operating within externally enforced correctness boundaries.

**RQ2:** Structural ratios show consistent increases in instruction count and memory operations, with modest shifts in control structures. These changes reflect a preference for correctness and stability over aggressive elimination. Despite structural expansion relative to Wyvern, the model maintains near-stable function counts, call behavior, and branching structure. Compared to the non-finetuned variant, it produces smaller binaries. This conservative profile illustrates how correctness constraints bias learning toward reliability-oriented transformation strategies. Importantly, this conservatism emerges from architectural coupling with verification, suggesting that governance mechanisms can shape generative optimization behavior toward predictable outcomes.

**RQ3:** Formal verification is central to maintaining semantic guarantees. For the 19 initially rejected programs, all converged to verified outputs within nine iterations, with a tight mean of approximately 5.3 and no divergence. Most failures were syntactic, indicating proximity to valid transformation boundaries. Verifier feedback therefore functions as a structured corrective signal that stabilizes generation rather than enabling open-ended exploration. The bounded convergence behavior further indicates that verification operates as a stabilizing controller within the pipeline, transforming generative rewriting into a controlled refinement process rather than an unbounded search.

**Fine-Tuning Enhances Optimization Stability:** The absence of extreme runtime and binary-size deviations suggests that fine-tuning improves calibration rather than aggressiveness. By internalizing verified patterns from Wyvern, the model exhibits consistent behavior across benchmarks and avoids erratic deviations. Fine-tuning acts as a regularizer, aligning transformations with formal correctness boundaries and promoting predictable optimization policies for production compiler pipelines. Taken together, these findings suggest that verified supervision and external enforcement serve as practical design principles for integrating generative AI into infrastructure software, where reliability, bounded behavior, and interpretability are essential.

## 7 Limitation and Future Work

While verification-guided learning shows promise, several constraints limit applicability. The study focuses on C programs over LLVM IR with Wyvern as the reference optimizer, restricting generalization to other languages, IRs, and architectures. Alive2 guarantees semantic equivalence but does not capture microarchitectural factors such as caching or branch prediction. Thus, verified transformations may exhibit hardware-dependent performance variability, exposing the gap between correctness and performance portability. Fine-tuning on Wyvern data introduces supervision bias, encouraging conservative behavior. Although this improves reliability, it may limit discovery beyond heuristic imitation. Repeated symbolic validation adds overhead, potentially affecting scalability.

Future work should explore performance-aware reinforcement learning with verifier-informed rewards, cross-IR evaluation, and lightweight verification techniques to assess whether verified learning can scale as a general architectural pattern for trustworthy AI-driven compiler optimization.

## 8 Conclusion

This work investigates whether a fine-tuned LLM can function as a compiler optimizer when correctness is enforced through formal verification. By integrating Alive2-based equivalence checking with feedback-guided regeneration, the system shows learned transformations can preserve semantic soundness while approximating a verified heuristic optimizer. The central contribution is architectural: correctness is enforced as a control mechanism rather than learned implicitly. This separation of generation and enforcement reframes verification as a runtime governance layer over generative behavior. Although the model does not surpass Wyvern in raw performance, it produces consistent correctness-preserving optimizations relative to a non-finetuned baseline. Verification constraints encourage conservative, reliability-oriented behavior for production toolchains. These findings demonstrate that generative AI components can be embedded within critical software infrastructure when bounded by deterministic validation and controlled feedback loops. More broadly, the results suggest foundation models can internalize semantics-sensitive transformation patterns under structured supervision, offering a scalable blueprint for trustworthy AI-driven compiler optimization. Beyond lazification, the verified learning architecture provides a reusable design pattern for integrating adaptive generative models into software engineering pipelines where correctness, stability, and deployment safety are essential.

## Acknowledgments

The authors used OpenAI's ChatGPT to assist with rephrasing and restructuring text for clarity. All research contributions, dataset construction, experimental design, and analysis were carried out solely by the authors.

## Data Availability

The dataset used in this paper is publicly available on Figshare[1]. The release includes the fine-tuning code and the complete set of experiments reported in the paper.

---

[1]https://figshare.com/s/0004d9a369cde01b0dce

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

Received 20 February 2007; revised 12 March 2009; accepted 5 June 2009

