# OpenReview forum: "Verified Learning for Compiler Optimization: An LLM-Guided Architecture with Formal Control"
_ACM.org/AIWare/2026/Conference — Submitted to AIware 2026_

### Official Review · Reviewer_NNNB · 2026-02-27

**Rating:** 2
**Confidence:** 4

**Review:**

The paper is well-written in general and easy to follow. However, I have the following concerns:

**Novelty:**
The paper presents limited novelty. In particular, the contribution LLM-VeriOpt [1] published in CGO 2026, presents and covers extensively the idea of using formal verification in order to train an LLM for the application of semantically correct and effective optimisations, in combination with formal verification. The authors of that paper claim a high percentage of success. However, there is no reference or comparison across LLM-VeriOpt and this work. As a result, the authors need to demonstrate how their approach is effective, novel and/or superior to that work and/or against other similar works.

**Related Work:**
Overall, the existing literature is briefly mentioned and not properly presented in comparison to this work. I highly recommend that the authors perform a comprehensive literature review, showcasing the novelty of their methodology in comparison to the literature, especially for works such as [1], and highlighting how they differentiate and how their contribution is novel and valuable.

**Methodology:**
Although the concept is straightforward, it is not clearly evident why you chose to focus on lazification, and what are the benefits over attempting to generate eagerly-executed optimizations. Please further elaborate about this decision. Ideally, a brief background section presenting it would be beneficial, accompanied with a list of benefits of focusing on it in your methodology.

**Effectiveness:**
While the authors claim that LLM-based optimisations can be successfully utilised, this is an overstatement. Given the low percentage rate of optimisations that led to faster results, I imply that significant work will be required to generate such optimisations. I would advise the authors to present potential areas of improvement, including potential reasons that the vast majority of optimisations performed worse that the analogous ones in Wyvern.

**Analysis of Results:**
While the authors present that 9.8% of the LLM-generated optimisations resulted in usable optimisations (i.e., resulting in better performance), it would be useful to analyze and demonstrate in more depth the RQ2 with focus on these successful cases, indicating common patterns and traits in the effective optimizations, as well as discussing potential reasons that these optimisations were better in comparison to those generated from Wyvern. This can give an indication of which specific direction the methodology of this work is effective. To this direction, Figure 4 should be refined and better presented as well.

**Availability:**
Although a results dataset is provided by the authors, no code repository related to the proposed methodology is available.
In addition, there are no instructions regarding how to use, reproduce, or interpret the provided data.

**Minor fixes:**
- Please fix the spacing in abstract and in text to preserve the padding in columns for lines 19 and 192, 335.
- You mention and utilise Meta Compiler LLM without a proper citation. Please revise.

[1] https://www.computer.org/csdl/proceedings-article/cgo/2026/11395239/2elc3bCHzFu

**Summary:**

The authors present a methodology in order to train an LLM to generate semantically correct and effective optimizations, focusing on  lazification in LLVM IR as a case study. The authors introduce C code snippets to LLVM CLang to generate unoptimized IR code, which then introduce in Wyvern optimizer to generate otimized IR. Then, they use the unoptimized/optimized IR pairs in order to train an LLM. Eventually, they use an LLM combined with the Alive2 formal verifier with the aim of generating semantically valid optimisations. The authors run experiments for the code of 8 well-known C projects, and demonstrated that, for their full set of experiments, about 9.8% of cases had better performance results while the optimized code was semantically valid with no violations.

---

> ### Author Response · Authors · 2026-03-20
> **Novelty**
>
> Thank you for the detailed and constructive feedback. We appreciate the concern about novelty, especially in relation to LLM-VeriOpt and other recent work. Our paper’s main contribution is not that it introduces formal verification into compiler optimization in the abstract, but that it studies a specific systems architecture in which generation and enforcement are cleanly separated at inference time. In our setting, the model is trained on verified Wyvern transformation pairs, but semantic correctness is not learned as part of training and is not trusted at deployment. Instead, Alive2 acts as an inline control layer that decides whether a candidate rewrite can enter the pipeline. This makes the central contribution architectural: the paper studies how a learned optimizer behaves when embedded inside a formally governed acceptance loop, and it analyzes the resulting conservatism, bounded convergence, and absence of semantic violations. We agree that this distinction should be stated more clearly when positioning the work relative to recent literature.

---

> ### Author Response · Authors · 2026-03-20
> **Methodology**
>
> We also appreciate the request for more explanation of why we chose lazification as the case study. We focused on lazification because it is a semantics-sensitive transformation whose profitability depends strongly on usage patterns and control-flow context. A parameter may be worth deferring when it is accessed conditionally or infrequently, but that same transformation can hurt performance if the value is reused heavily, because thunk creation and memoization introduce overhead. This makes lazification a useful test case for our question: can a learned model participate in a transformation setting where correctness must be guaranteed and profitability is context-dependent? In that sense, lazification is not just a convenient example. It is a demanding setting where both the decision to transform and the shape of the rewrite matter. The paper already emphasizes this tension in the introduction, the problem formulation, and the description of Wyvern’s profitability model.

---

> ### Author Response · Authors · 2026-03-20
> **Effectiveness**
>
> On the concern that our effectiveness claim may be overstated, thank you for pointing this out. We agree that the contribution should not be interpreted as broad performance superiority over Wyvern. The empirical picture in the paper is more measured than that. Wyvern remains faster on most benchmarks, and the fine-tuned model is clearly more conservative, applying the same or fewer lazifications and avoiding over-lazification altogether. The paper reports that about 9.8% of benchmarks achieve comparable or improved runtime under the learned system, while roughly 76% show fewer transformations with slightly slower runtime. For that reason, the strongest claim of the paper is not that the learned system outperforms the heuristic baseline, but that it can participate safely in the optimization loop under formal control, with bounded verification overhead and stable convergence when regeneration is needed. We appreciate this comment and agree that this framing should be made more precise.

---

> ### Author Response · Authors · 2026-03-20
> **Analysis of Results**
>
> For the request to analyze the successful 9.8% cases in more depth and to expand the discussion of why those cases outperform Wyvern, thank you for the helpful suggestion. We agree that this would be valuable, and we will consider it as future work.

---

> ### Author Response · Authors · 2026-03-20
> **Availability**
>
> Regarding availability and reproducibility, thank you for raising this concern. We would like to clarify that the anonymous artifact link already includes all code written and used in this study, including the fine-tuning and experimental pipeline used to produce the reported results. In addition, the paper references both the specific model used in our experiments and the Wyvern tool on which the training data generation is based. We apologize if this was not sufficiently clear in the current manuscript. In the revision, we will make the contents of the anonymous release and the reproduction pathway more explicit so that readers can more easily identify the provided artifacts and understand how they connect to the results reported in the paper.

---

### Official Review · Reviewer_WNZ2 · 2026-02-28

**Rating:** 2
**Confidence:** 4

**Review:**

## Strengths

+ Critical topic: The integration of formal verification to guarantee 100% semantic preservation directly tackles the most significant barrier to deploying LLMs in production compiler toolchains, i.e., trustworthiness.

+ Effective feedback: The closed-loop design utilizing SMT-based counterexamples for self-correction is well-engineered.

## Weaknesses

- Supervision bias: The LLM is fine-tuned exclusively on transformations generated by an existing heuristic optimizer (Wyvern). Consequently, the model suffers from severe supervision bias. It essentially learns to safely imitate a traditional tool rather than discovering novel optimization patterns. This architectural choice inherently caps the model's performance ceiling.

- Generalization: The evaluation is restricted to a single, relatively localized optimization pass (lazification). It remains highly questionable whether this LLM-guided architecture can generalize to more complex, global optimizations, where the search space is exponentially larger.

- Minor improvement: Even the evaluation is limited to lazification, the reported performance gain is modest. This raises concerns about the practical impact of the proposed approach, especially considering the additional complexity and overhead introduced by the LLM integration.

## Questions

1. Why is the model not trained on a broader dataset of human-written, highly optimized code (e.g., from competitive programming or performance-critical libraries) to encourage the discovery of novel optimizations beyond the baseline's heuristics?

2. Have the authors considered exploring methods to empower the LLM to generate more creative transformations, rather than relying solely on supervised fine-tuning?

3. Can the authors provide more evaluation results on more diverse optimization scenarios to demonstrate the generalization capabilities of their approach?

**Summary:**

This paper introduces a framework that uses an LLM to propose compiler optimizations while relying on Alive2 formal verification to guarantee correctness. The model is fine-tuned on lazification transformations generated by Wyvern and, during inference, must regenerate any rewrite that fails equivalence checking. Experiments on the LLVM test suite show that the LLM reproduces most of Wyvern’s behavior, applies fewer and more conservative transformations, achieves performance comparable or better on about 10% of benchmarks, and produces zero semantic errors. The work demonstrates that LLMs can safely participate in compiler optimization when governed by formal verification.

---

> ### Author Response · Authors · 2026-03-20
>
> Thank you for the encouraging comments about the trustworthiness focus and the closed-loop design. We appreciate the questions regarding broader training data, more creative transformation strategies, and evaluation on more diverse optimization scenarios. These are all important directions for extending the current study. In the present paper, our goal was to test a verification-centered architecture in a controlled setting where the transformation space is semantics-sensitive and every accepted rewrite can be checked mechanically. We agree that broader datasets, more exploratory training strategies, and additional optimization domains would strengthen the generality of the framework, and we will consider these directions in future work. The current paper already acknowledges supervision bias and limited generalization scope as core limitations of the present study.

---

### Official Review · Reviewer_cndM · 2026-03-09

**Rating:** 2
**Confidence:** 3

**Review:**

Strengths
- The motivation for the work is strong. It would be great if we can replace complex, hand-engineered systems with something fundamentally simpler, like a neural network, while still maintaining comparable overall performance. For example, Tesla claims to have recently replaced 300K LoC of C++ code with simpler and more performant neural networks (this might be a nice citation to add to the work :))
- The evaluation provides useful insight into how the learned model behaves

Weaknesses
- The fine-tuning approach is very simple. It completely depends on the pre-existing optimizer to learn how to optimize.
- Given that the approach is entirely dependent on the pre-existing optimizer, it is unlikely the LLM will ever match the pre-existing optimizer's performance, and indeed this is what the author's results show
- Lack of comparison to simple baselines (e.g. simple use Anthropic, OpenAI, or Google LLMs out of the box to propose optimizations)

Other minor feedbacks:
- Line 74 -- [19] is not the correct citation for MLGO
- Minor: Line 113 -- Importantly, correctness is not learned by the model but enforced externally by the verifier. — I don’t think this is a positive. Ideally the model has some notion of correctness so it proposes fewer incorrect optimizations.
- Minor: The paper could do with a short, concrete example of a lazification optimization in LLVM

Since their lacks any novelty in the proposed approach, the overall result is negative, and the evaluation is limited, I would put this paper at slightly below the bar for workshop acceptance. If any one of these were addressed I would recommend acceptance.

**Summary:**

The authors propose to train an LLM to learn compiler optimizations by generating training examples using existing, hand-engineered optimizers, and then enforce correctness of the LLM’s optimizations at compile time using formal verification. The authors present a case study learning lazification optimizations using Wyrven, a rule-based lazification optimizer, and use Alive2 for formal verification. The authors compare their learned optimizer with Wyrven, and a non-fine-tuned baseline LLM.

---

> ### Author Response · Authors · 2026-03-20
>
> Thank you for the thoughtful feedback and for recognizing the motivation and the behavioral analysis as strengths of the paper. On the point about correctness being enforced externally rather than learned by the model, we agree that this statement needs clearer framing. Our intent was not to suggest that it is better for the model to have no notion of correctness. Rather, the point is architectural: in our system, semantic correctness does not depend on the model being reliable in every case, because Alive2 serves as the final acceptance mechanism for every rewrite. This separation lets the model act as a proposal generator while the verifier remains the authority on semantic preservation. In other words, the guarantee comes from the system design, not from assuming that the model will always internalize correctness perfectly. This is especially important in compiler settings, where even rare semantic mistakes are unacceptable. The paper already reflects this design choice in both the introduction and approach sections, where transformation generation and correctness enforcement are deliberately separated.

---

> ### Author Response · Authors · 2026-03-20
>
> We also appreciate the suggestion to include a short concrete example of lazification. In this work, lazification refers to converting eager call-by-value computation into deferred call-by-need execution when that change is likely to be profitable. A simple example is a function argument whose value is computed eagerly but is only used inside a conditional branch or is rarely accessed along common execution paths. In such a case, the compiler can replace immediate evaluation with a thunk and memoization logic so the value is computed only if it is actually needed. This is exactly the type of transformation Wyvern performs to generate the training pairs used in our study. We agree that making this example explicit would improve readability, especially for readers who are less familiar with lazification in LLVM-based optimization.
>
> For the suggestion about adding comparisons to out-of-the-box proprietary LLMs, thank you for raising this point. We agree that this would be valuable, and we will consider it in future work.